# DNA damage shifts circadian clock time via Hausp-dependent Cry1 stabilization

Stephanie J Papp[†], Anne-Laure Huber[†], Sabine D Jordan, Anna Kriebs, Madelena Nguyen, James J Moresco, John R Yates III, Katja A Lamia*

Department of Chemical Physiology, Scripps Research Institute, La Jolla, United States

**Abstract** The circadian transcriptional repressors cryptochrome 1 (Cry1) and 2 (Cry2) evolved from photolyases, bacterial light-activated DNA repair enzymes. In this study, we report that while they have lost DNA repair activity, Cry1/2 adapted to protect genomic integrity by responding to DNA damage through posttranslational modification and coordinating the downstream transcriptional response. We demonstrate that genotoxic stress stimulates Cry1 phosphorylation and its deubiquitination by Herpes virus associated ubiquitin-specific protease (Hausp, a.k.a Usp7), stabilizing Cry1 and shifting circadian clock time. DNA damage also increases Cry2 interaction with Fbxl3, destabilizing Cry2. Thus, genotoxic stress increases the Cry1/Cry2 ratio, suggesting distinct functions for Cry1 and Cry2 following DNA damage. Indeed, the transcriptional response to genotoxic stress is enhanced in $Cry1^{-/-}$ and blunted in $Cry2^{-/-}$ cells. Furthermore, $Cry2^{-/-}$ cells accumulate damaged DNA. These results suggest that Cry1 and Cry2, which evolved from DNA repair enzymes, protect genomic integrity via coordinated transcriptional regulation.

*For correspondence: klamia@scripps.edu

[†]These authors contributed equally to this work

Competing interests: The authors declare that no competing interests exist.

## Introduction

Mammalian circadian clocks involve transcriptional feedback loops (*Green et al., 2008*): Brain and muscle ARNT-like protein 1 (BMAL1) and 'circadian locomotor output cycles kaput' (CLOCK) activate expression of many transcripts including the period (*Per1*, *Per2*, and *Per3*) and cryptochrome (*Cry1* and *Cry2*) genes, whose protein products (PERs and CRYs) inhibit CLOCK and BMAL1, resulting in rhythmic expression. Posttranslational modifications reset the clock (*Green et al., 2008*), including ubiquitination and subsequent degradation of CRYs by Skp-Cullin-Fbox (SCF) E3 ligases in which substrates are recruited by F-box and leucine-rich repeat proteins 3 (FBXL3) (*Busino et al., 2007*; *Siepka et al., 2007*) and 21 (FBXL21) (*Dardente et al., 2008*; *Hirano et al., 2013*; *Yoo et al., 2013*). Phosphorylation of CRY1 by AMP-activated protein kinase (AMPK) increases its association with FBXL3 (*Lamia et al., 2009*) by disrupting interaction with PER (*Xing et al., 2013*). CRY stability seems to be a key factor in circadian period determination: several mutants identified in forward genetic screens selected by robust changes in circadian period have been alleles of FBXL3 or FBXL21 (*Godinho et al., 2007*; *Siepka et al., 2007*; *Yoo et al., 2013*).

In addition to their roles in circadian clock negative feedback, Cry1 and Cry2 are key effectors of a variety of physiological pathways. In mammals, Cry1 and Cry2 modulate glucose homeostasis by repressing the transcriptional activity of the glucocorticoid receptor (*Lamia et al., 2011*) and the CRE-responsive element binding protein (CREB) (*Zhang et al., 2010*). Consistent with these results, small molecules that stabilize Cry1/2 depress glucose production in hepatocytes and may be useful in the treatment of hyperglycemia (*Hirota et al., 2012*). Genetic disruption of both Cry1 and Cry2 also alters the expression of proinflammatory cytokines (*Narasimamurthy et al., 2012*), the severity of arthritis (*Hashiramoto et al., 2010*), and salt-induced blood pressure elevation (*Doi et al., 2010*). Genetic inactivation of Cry1 and/or Cry2 has also been reported to alter rates of tumor formation

**eLife digest** Many aspects of our physiology and behavior, most notably our patterns of sleep and wakefulness, are synchronized with the day–night cycle. These circadian rhythms are generated and maintained by the circadian clock, which consists of positive and negative feedback loops formed by a large number of genes and proteins. The end result is that the rates at which thousands of proteins are produced varies rhythmically over the course of the day–night cycle.

It has long been suspected that one of the functions of this circadian clock is to control the timing of cell division. Moreover, since UV radiation can give rise to genetic mutations when cells divide, it is thought that the circadian clock limits the amount of DNA damage that occurs during daytime. Papp, Huber et al. have now confirmed that the circadian clock does indeed participate in the DNA damage response and have revealed that two proteins known to be involved in the circadian clock- —*Cryptochrome 1* and *2*—have a central role in protecting the integrity of the genetic information in the cell. These proteins evolved from light-activated enzymes that repair DNA in bacteria.

While mammalian cryptochromes have lost their ability to repair DNA, they still prefer to bind to genetic material that has been damaged by UV radiation. Papp, Huber et al. show that DNA damage triggers cryptochrome 1 to bind to a protein called Hausp, which stabilizes the cryptochrome and prevents it from being broken down. By contrast, DNA damage triggers cryptochrome 2 to bind to a protein called Fbxl3, which has a destabilizing effect on the cryptochrome and promotes its degradation. Since the cryptochromes regulate the activity of BMAL1 and CLOCK, the proteins associated with the two master clock genes, these changes can have a significant effect on the circadian clock of an organism.

Further experiments are needed to work out how these proteins influence the activity of BMAL1 and CLOCK, and to investigate the seemingly conflicting roles of the two cryptochromes and the interactions between them.

(*Ozturk et al., 2009*), though the reported effects have varied (*Fu and Kettner, 2013*). In addition, Cry-deficient mice are resistant to genotoxic stress in the context of cyclophosphamide treatment (*Gorbacheva et al., 2005*). Consistent with the idea that Cry1/2 may be promiscuous transcriptional repressors involved in a wide variety of physiological pathways, a recent study found that Cry1 and Cry2 each bound thousands of chromatin sites independently of other clock transcription factors in mouse liver (*Koike et al., 2012*).

Though Cry1 and Cry2 are mostly believed to associate with chromatin via binding a variety of transcription factors, they can also interact directly with DNA. Cry1 and Cry2 evolved from prokaryotic light-activated DNA repair enzymes, known as photolyases. While they seem to have lost the [6-4] photolyase catalytic activity characteristic of their ancestral homologs (*Ozturk et al., 2007*), they retain the ability to bind preferentially to UV-damaged DNA containing a [6-4]photoproduct (*Ozgur and Sancar, 2003*). The three-dimensional structures of Cry1 and Cry2 resemble those of photolyases, including the DNA binding surfaces (*Maul et al., 2008*; *Czarna et al., 2013*; *Xing et al., 2013*). Together, these properties suggest that Cry1 and Cry2 could retain a residual role in sensing or responding to damaged DNA. Such conservation of function by divergent molecular mechanisms has been seen previously between cryptochromes derived from different species (*Yuan et al., 2007*; *Lamia et al., 2009*; *Kim et al., 2014*).

Ubiquitination of substrate proteins by E3 ligases, like $SCF^{Fbxl3}$ and $SCF^{Fbxl21}$, is reversed by ubiquitin-specific proteases (USPs) (*Eletr and Wilkinson, 2014*). Herpes virus associated ubiquitin-specific protease (Hausp; a.k.a. Usp7) was first identified as the cellular partner of the herpes virus protein Vmw110 (*Everett et al., 1997*). Hausp modulates proliferation by catalyzing the removal of polyubiquitin chains from the tumor suppressor p53 and from the p53-destabilizing E3 ligases Mdm2 and MdmX (*Li et al., 2002*, *2004*). The affinity of Hausp for p53 is increased and for Mdm2/MdmX is decreased in response to DNA damage (*Khoronenkova et al., 2012*), contributing to stabilization of p53. Knockout of Hausp in mice is lethal (*Kon et al., 2010*), probably due to disrupted cell proliferation. A growing list of Hausp substrates has been identified recently, including several components of DNA damage response and DNA repair pathways (*Nicholson and Suresh Kumar, 2011*; *Schwertman et al., 2012*; *Jacq et al., 2013*; *Eletr and Wilkinson, 2014*). In this study, we

demonstrate that Hausp participates in DNA damage-induced resetting of circadian clock time by stabilizing Cry1.

## Results

### Identification of Hausp as a novel regulator of Cry1 stability

In an ongoing effort to understand the molecular determinants of cryptochrome stability, we used mass spectrometry to identify novel protein partners of mammalian CRYs and found Hausp to be the most highly enriched protein in Cry1-containing complexes (*Figure 1A–B*, *Supplementary file 1*). Co-immunoprecipitation of endogenous (*Figure 1C*) and overexpressed (*Figure 1D–E*) Cry1 and Hausp confirmed the specificity of this interaction. Interestingly, Hausp interacts much more strongly with Cry1 than with the closely related Cry2 (*Figure 1D*). Indeed, the divergent Cry1 C-terminus is necessary and sufficient for strong interaction with Hausp (*Figure 1D–E*).

The regulation of Cry1/2 protein stability is complex and involves differential expression and localization of the E3 ligase subunits Fbxl3 and Fbxl21 that compete for Cry binding and have different rates of ubiquitin conjugation (*Hirano et al., 2013*; *Yoo et al., 2013*). Similar to what has been described for Fbxl3 and Fbxl21, we found no significant tissue specificity or circadian rhythm of expression or localization for Hausp (*Figure 1—figure supplements 1, 2*). However, while both Hausp and Cry1 are more abundant in the cytoplasm, their interaction is stronger in the nucleus, regardless of circadian phase (*Figure 1C*, Circadian Time, CT, denotes hours after dexamethasone-induced synchronization of circadian cycles).

Because Hausp is an ubiquitin-specific protease, its interaction with Cry1 in the nucleus seemed likely to stabilize nuclear Cry1 by removing polyubiquitin chains. We used small hairpin RNA (shRNA)-expressing viruses to demonstrate that Hausp depletion led to decreased Cry1 protein primarily in the nuclear compartment in mouse embryonic fibroblasts (MEFs) independent of circadian phase, as expected from the ubiquity of Hausp expression (*Figure 2A*, *Figure 2—figure supplement 1*). Treatment of cells with pharmacological inhibitors of Hausp (*Nicholson and Suresh Kumar, 2011*; *Weinstock et al., 2012*) also decreases Cry1 protein, especially in the nucleus (*Figure 2B*), consistent with the hypothesis that Hausp stabilizes nuclear Cry1 in vivo. (Note that compound 7 also inhibits Usp47.)

Recombinant Hausp can deubiquitinate Cry1 in vitro (*Figure 2—figure supplement 2*). To examine Cry1 ubiquitination in vivo, we measured ubiquitinated Cry1 in MEFs expressing control or Hausp-targeting shRNA in the presence or absence of the proteasome inhibitor MG132 to stabilize ubiquitinated proteins. Cry1 from Hausp-depleted cells was much more highly ubiquitinated than Cry1 from control cells (*Figure 2C*, *Figure 2—figure supplement 3*) as expected if Hausp catalyzes the removal of polyubiquitin chains from Cry1 in vivo. (Note that while Cry1 is decreased in Hausp-depleted cells, we used a limiting amount of anti-Cry1 antibody for immunoprecipitation to normalize the amount of Cry1 and enable comparison between samples. $Cry^{-/-}$ cells were used as a control to subtract the non-specific ubiquitin signal and enable quantitative comparison of control and Hausp-depleted cells.)

### Hausp activity alters circadian period length

Cryptochrome stability is a critical determinant of circadian period length, though the direction and magnitude of the period change associated with altered expression or stability of Cry1 and/or Cry2 seems to depend on the mechanism and context of altered stability (*Vitaterna et al., 1999*; *Hirota et al., 2012*; *St John et al., 2014*). Nonetheless, if Hausp stabilizes Cry1 by removing ubiquitin chains, reducing Hausp expression or activity is expected to alter circadian rhythms. In immortalized fibroblasts expressing a Per2-Luciferase fusion from the endogenous Per2 locus (*Per2::Luc* [*Yoo et al., 2004*]), shRNA-mediated depletion of Hausp increased circadian period (*Figure 2D,E*). We also observed period lengthening in immortalized *Per2::Luc* MEFs when Hausp activity was inhibited pharmacologically (*Figure 2F,G*). Because our data suggest that Hausp inhibition and AMPK activation each destabilizes nuclear Cry1, we examined whether they could synergistically increase circadian period. Using cells stably expressing luciferase under a circadian promoter (*U2OS-B6* [*Vollmers et al., 2008*]), we observed that activation of AMPK increased the circadian period as expected (*Lamia et al., 2009*), inhibition of Hausp also increased period, and combined activation of AMPK and inhibition of Hausp led to a dramatic increase in period, perhaps reflecting synergistic destabilization of nuclear Cry1 (*Figure 2H,I*).

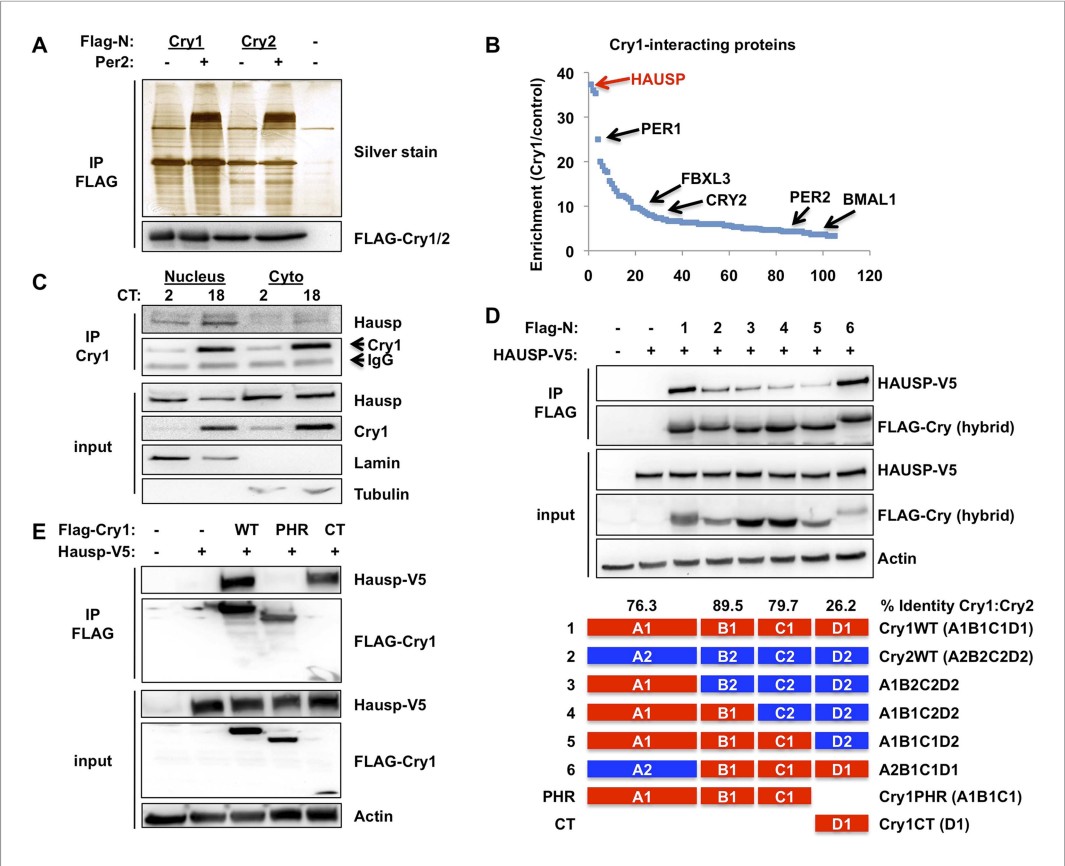

**Figure 1**. Hausp interacts with Cry1. (**A** and **B**) Lysates from 293T cells expressing pcDNA3-2xFLAG with no insert (–), Cry1, or Cry2 after the FLAG tag with (+) or without (–) co-expression of Per2 were used to purify control, Cry1, or Cry2-containing complexes by immunoprecipitation (IP) of the FLAG tag. 5% of each purification was analyzed by SDS-PAGE and silver stain (**A**) and components of the resulting complexes were identified by mass spectrometry performed on the remaining 95% of the sample. The experiment was performed in triplicate and Pattern Lab for Proteomics (*Carvalho et al., 2012*) was used to identify statistically enriched partners. In (**B**) Enrichment is the ratio of spectral counts in Cry1 vs control samples for all statistically enriched partners over three experiments (e.g., lane 1 vs lane 5 from [**A**]). Arrows depict several established partners for Cry1 as well as the observed 37-fold enrichment for Hausp in Cry1-containing samples. (**C**) Endogenous Hausp bound to endogenous Cry1 was detected by immunoblot (IB) following IP from nuclear and cytoplasmic fractions of mouse embryonic fibroblasts (MEFs) harvested at the indicated times (CT, hours) following circadian synchronization by dexamethasone. (**D**) Top: Hausp-V5 bound to FLAG-Cry1/2 hybrids was detected by IB following IP from 293T cells. Bottom: schematic diagram showing the composition of the Cry1/2 hybrids and domains used in **D** and **E**. (**E**) Hausp-V5 bound to FLAG-Cry1 full length or isolated domains was detected by IB following IP.

The following figure supplements are available for figure 1:

**Figure supplement 1**. Circadian measurement of *Hausp* mRNA expression in mouse tissues.

**Figure supplement 2**. Circadian measurement of Hausp protein expression in mouse tissues.

## DNA damage increases the Cry1/Cry2 ratio

Given that the Cry1–Hausp interaction occurs primarily in the nucleus and that Hausp interaction with other partners is regulated by DNA damage, we examined the impact of DNA damage on the Hausp–Cry1 association and found that it increases the interaction (*Figure 3A*, *Figure 3—figure supplements 1, 2*). Because Hausp catalyzes the removal of polyubiquitin chains from Cry1 thereby decreasing its proteasomal degradation (*Figure 2*), increased Cry1–Hausp association in response to genotoxic stress leads to a prediction that DNA damage should increase Cry1 protein levels.

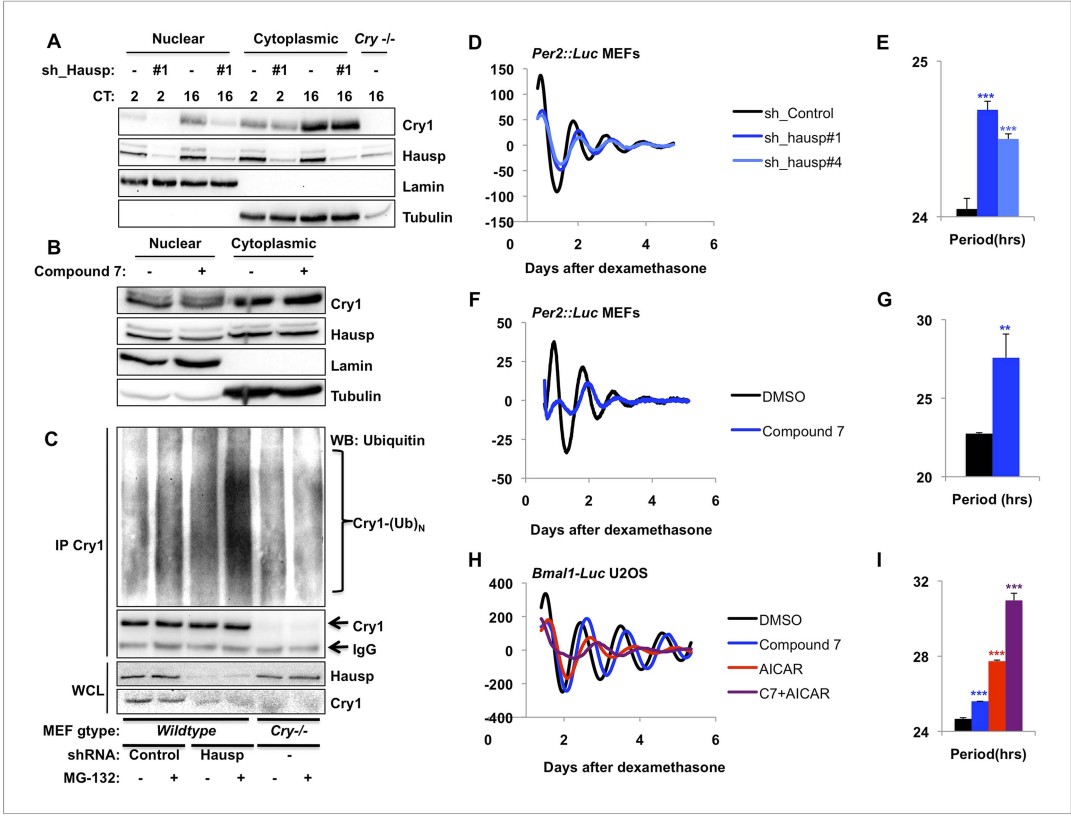

**Figure 2**. Hausp stabilizes Cry1 via deubiquitination and alters circadian rhythms. (**A**) Wild-type or $Cry1^{-/-};Cry2^{-/-}$ ($Cry^{-/-}$) MEFs stably expressing a control sequence (−) or shRNA targeting Hausp (#1) were subjected to nuclear and cytoplasmic fractionation. Cry1, Hausp, Lamin, and Tubulin were analyzed by IB from fractions harvested at the indicated times following circadian synchronization with dexamethasone (CT, hours). (**B**) Cry1, Hausp, Lamin and Tubulin were detected by IB in nuclear and cytoplasmic fractions from MEFs treated with vehicle (DMSO, −) or Compound 7 (+). (**C**) Wild-type MEFs stably expressing control or Hausp-targeting shRNA or $Cry^{-/-}$ MEFs were treated with vehicle (DMSO, −) or MG132 (+) for 6 hr, and lysed in RIPA buffer containing iodoacetamide. 6 mg of RIPA lysates from each condition was subjected to IP with 5 µg of anti-Cry1 antibody. Ubiquitinated Cry1 (Cry1–(Ub)$_N$), Cry1, and Hausp were detected by IB in IPs and whole cell lysates (WCL). (**D**, **F**, **H**) Typical results of continuous monitoring of luciferase activity from MEFs expressing Per2-luciferase fusion protein from a knock-in allele (**D** and **F**) or from U2OS cells stably expressing luciferase under the control of the Bmal1 promoter (**H**) with stable expression of control or either of two shRNA sequences targeting Hausp (**D**) or treated with Compound 7 and/or AICAR (**F** and **H**). (**E**, **G**, **I**) Quantitation of the circadian period of luciferase activity from experiments performed as described in (**D**, **F**, **H**). Data represent the mean ± s.d. for 4–8 samples per condition. **p < 0.01, ***p < 0.001 vs control samples (control shRNA for **E** or DMSO-treated cells for **G** and **I**).

The following figure supplements are available for figure 2:

**Figure supplement 1**. Validation of shRNA targeting Hausp.

**Figure supplement 2**. In vitro deubiquitination of Cry1 by recombinant Hausp.

**Figure supplement 3**. Quantitation of in vivo Cry1 ubiquitination.

Consistent with this hypothesis, we found that exposure to DNA damage transiently stabilized endogenous Cry1 in primary MEFs (*Figure 3A–C*). Intriguingly, Cry2 was destabilized following exposure to DNA damage, demonstrating that the increase in Cry1 does not merely reflect a change or synchronization of the circadian rhythm and suggesting differential regulation of these highly homologous family members, consistent with our observation that Hausp preferentially interacts with Cry1. Because Cry1 and Cry2 each can repress the other's expression, Cry2 protein could decrease in response to damage secondary to stabilization of Cry1. However, Cry2 protein decreases and Cry1

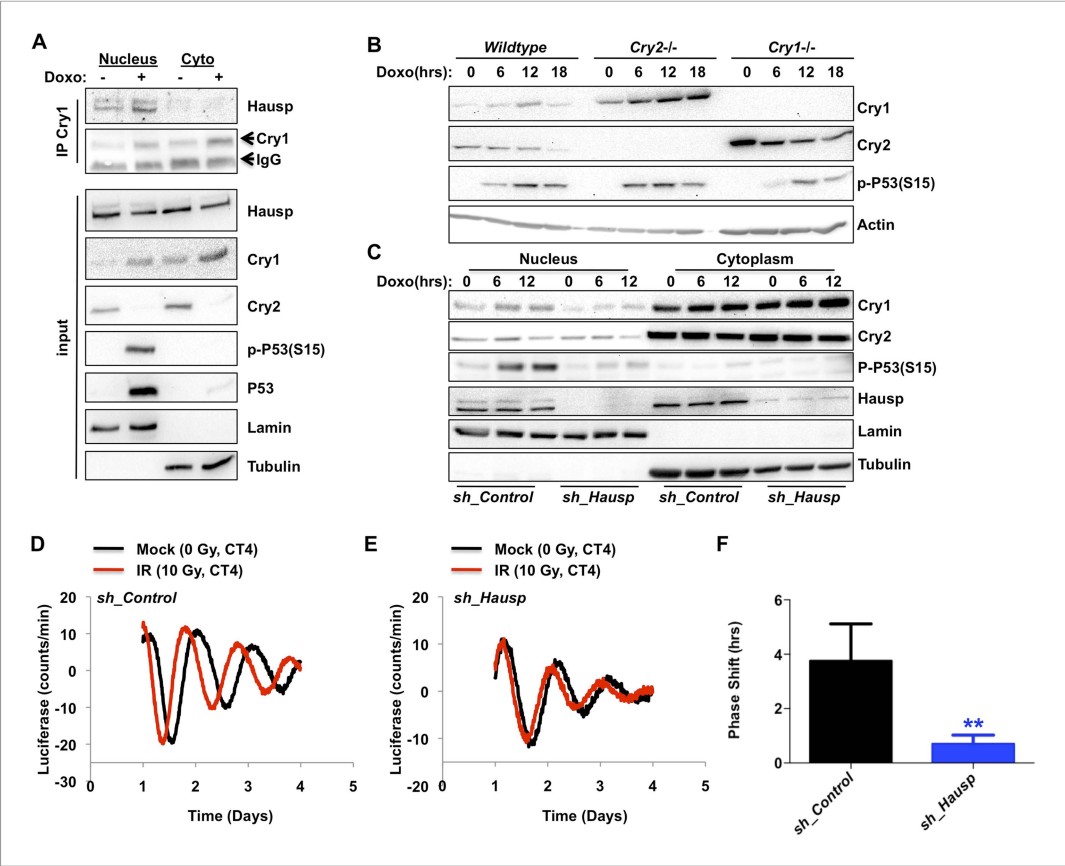

**Figure 3**. DNA damage resets the clock via Hausp-dependent stabilization of nuclear Cry1. (**A**) Endogenous Hausp, Cry1, Cry2, phospho-P53 (Ser15), P53, Lamin, and Tubulin were detected by IB in Cry1 immunoprecipitates or input samples from nuclear and cytoplasmic fractions of primary MEFs treated with vehicle (−) or doxorubicin (+). (**B**) Cry1, Cry2, phospho-P53 (Ser15), and Actin were detected by IB in lysates from wildtype (*WT*), *Cry1*$^{-/-}$ or *Cry2*$^{-/-}$ MEFs treated with doxorubicin for the indicated times. (**C**) Cry1, Cry2, phospho-P53 (Ser15), Hausp, Lamin, and Tubulin were detected by IB in nuclear and cytoplasmic fractions from MEFs expressing control or Hausp-targeting shRNA and treated with doxorubicin for the indicated times. (**D** and **E**) Typical results of continuous monitoring of luciferase activity from primary adult ear fibroblasts expressing Bmal1-luciferase and control or Hausp-targeting shRNA and treated with 0 (black) or 10 Gy (red) irradiation 3 hr after circadian synchronization with dexamethasone. Data represent the mean luciferase counts of eight samples per condition from one of four independent experiments. (**F**) Quantitation of the differences in initial circadian phase of luciferase activity caused by irradiation calculated from experiments performed as described in (**D** and **E**). Data in (**D**–**F**) represent the mean ± s.d. of phase shifts observed in four independent experiments. **p < 0.01 vs control samples.

The following figure supplements are available for figure 3:

**Figure supplement 1**. Effect of DNA damage on Cry1-Hausp interaction in transfected 293T cells.

**Figure supplement 2**. Proteostasis and/or membrane stress increase the Cry1-Hausp interaction.

**Figure supplement 3**. Circadian time of exposure determines phase shift in response to DNA damage.

protein increases in response to DNA damage in MEFs expressing only a single Cry paralog (i.e., Cry2 in *Cry1*$^{-/-}$ MEFs and vice versa; *Figure 3B*). Thus, DNA damage acutely regulates Cry1 and Cry2 protein levels independently.

To determine the contribution of Hausp to the damage-induced stabilization of nuclear Cry1, we examined nuclear Cry1 protein levels following DNA damage in MEFs expressing either control sequences or Hausp-targeting shRNA. Indeed, depletion of Hausp prevents damage-induced stabilization of nuclear Cry1, similar to the effect of Hausp depletion on p53 accumulation

(*Figure 3C*). Note that the weaker interactions that we observed between Hausp and Cry2 or hybrid constructs containing the C-terminus of Cry2 compared to those containing the Cry1 C-terminus (*Figure 1D*) are likely artefacts of overexpression in 293T cells since we did not observe destabilization of Cry2 upon Hausp depletion (*Figure 3C*).

## Hausp is required for clock resetting in response to DNA damage

It has been reported that DNA damage causes phase shifts of circadian rhythms (*Oklejewicz et al., 2008*; *Engelen et al., 2013*). Consistently, we observed phase shifts in primary MEFs with a peak shift following irradiation at CT2-4, (*Figure 3—figure supplement 3*). The requirement for Hausp in stabilization of nuclear Cry1 after DNA damage suggested Hausp could contribute to phase shifts in response to DNA damage. By examining the circadian phase of control and Hausp-depleted fibroblasts after exposure to irradiation at CT3, we found that although the circadian phase of the non-irradiated cells is similar (*Figure 2D*), DNA damage-induced phase shifts were greatly diminished in Hausp-deficient fibroblasts (*Figure 3D–F*).

## Phosphorylation of both partners influences the association between Cry1 and Hausp

ATM- and PPM1G-dependent dephosphorylation of serine 18 in the N-terminus of Hausp has been reported to drive the DNA damage dependent disruption of Hausp interaction with Mdm2 and MdmX (*Khoronenkova et al., 2012*). Conversely, S18 de-phosphorylation may increase Hausp–Cry1 association because mutation of S18 to the non-phosphorylatable amino acid alanine (S18A) increases interaction and mutation to aspartic acid, which is chemically similar to phospho-serine, decreases the interaction (*Figure 4A*). However, S18 dephosphorylation cannot fully explain DNA damage induction of Cry1–Hausp interaction as evidenced by persistent stimulated association between Cry1 and Hausp S18A after DNA damage. Intriguingly, we (*Figure 4—figure supplement 1*, *Supplementary file 2*) and others (*Gao et al., 2013*) find that Cry1 and Cry2 interact with kinases that are activated by DNA damage and phosphorylate serine or threonine followed by glutamine, (S/T)-Q (*Kim et al., 1999*; *O'Neill et al., 2000*). Cry1 and Cry2 contain several such sequences (*Figure 4—figure supplement 2*), including three serines in the Cry1 C-terminal tail that are not conserved in Cry2.

Using an antibody that recognizes phospho-(S/T)Q, we determined that Cry1 is rapidly phosphorylated in response to either chemically or radiation-induced DNA damage, while damage-induced phosphorylation of Cry2 on (S/T)Q was reduced and delayed compared to that of Cry1 (*Figure 4B* and not shown), indicating that the rapid phosphorylation of Cry1 in response to DNA damage likely occurs on the non-conserved C-terminal tail. Mutating each or all of the Cry1 C-terminal SQ serines to alanine decreased or abolished, respectively, the phospho-(S/T)Q signal after DNA damage, indicating that these are the sites in Cry1 that are rapidly phosphorylated in response to DNA damage (*Figure 4C*, and not shown). Notably, S588 is the only one of these sites on which phosphorylation has been directly detected in vivo (*Lamia et al., 2009*; *Hegemann et al., 2011*). We generated an antibody that specifically recognizes Cry1 phosphorylated on S588 and measured a rapid increase in the presence of this phosphorylated species after exposure to DNA damage (*Figure 4D*). Consistent with the reported stabilization of Cry1 by mimicking phosphorylation at S588, mutation of this site to aspartic acid increased its association with Hausp (*Figure 4E*, left).

## DNA damage stimulates the association of Cry1/2 with Fbxl3

Because the effects of DNA damage on Cry1 and Cry2 stability are not fully explained by regulated interaction with Hausp, we examined the effect of DNA damage and subsequent phosphorylation events on the interactions between Cry1/2 and Fbxl3. To our surprise, prolonged exposure to DNA damage increases the interactions of Cry1 and especially Cry2 with Fbxl3 (*Figure 4F*), probably contributing to the transient nature of the Cry1 stabilization and to Cry2 destabilization following damage. Notably, mutation of Cry1 S588 to aspartic acid, which increases Cry1 interaction with Hausp, decreases the association of Cry1 with Fbxl3 (*Figure 4E*, right) suggesting that phosphorylation of the unique Cry1 C-terminus may oppose the increase in Fbxl3 binding to Cry1, possibly explaining the preferential induction of Fbxl3 binding to Cry2 compared to Cry1.

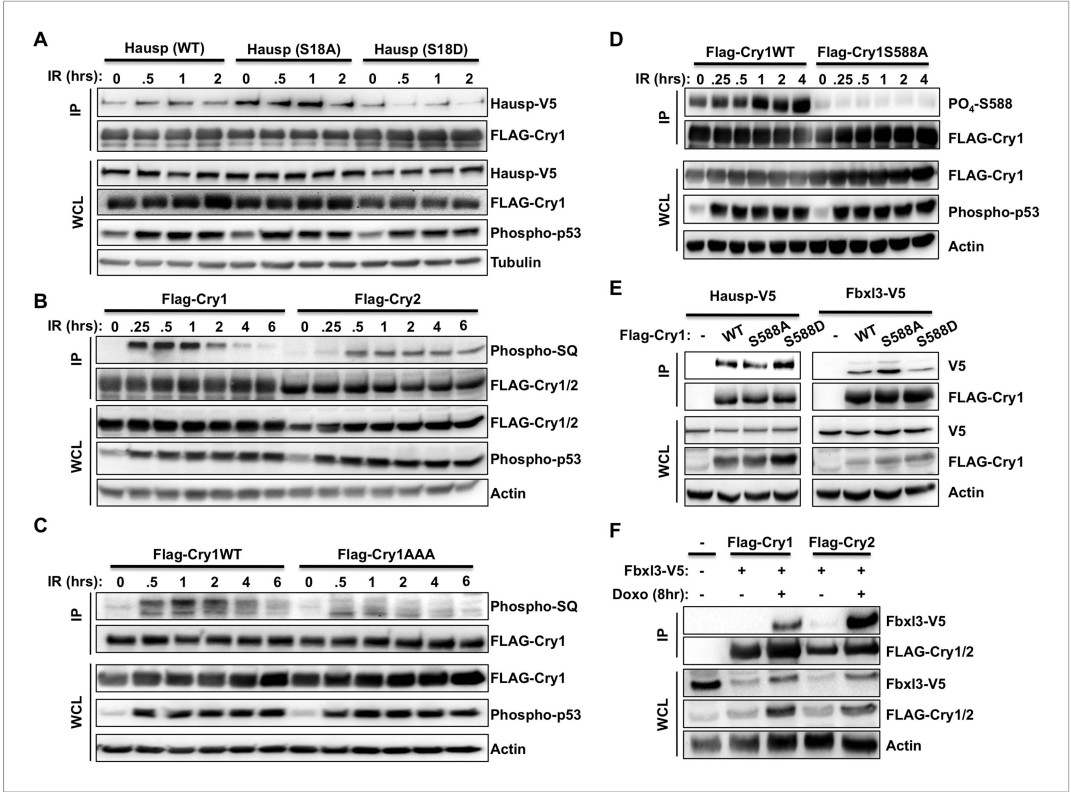

**Figure 4**. DNA damage induced signaling modulates interactions of Cry1/2, Hausp, and Fbxl3. Hausp-V5, FLAG-Cry1/2, phospho-P53 (Ser15), Phospho-SQ/TQ, Phospho-Cry1S588 (P-S588), Fbxl3-V5, and Actin were detected by IB in IPs and lysates (WCL) from 293T cells transfected with the indicated plasmids and lysed at the indicated times following treatment with doxorubicin (doxo) or irradiation (IR).

The following figure supplements are available for figure 4:

**Figure supplement 1**. Composition of Cry1- and Cry2-associated protein complexes.

**Figure supplement 2**. Conserved SQ/TQ motifs present in Cry1 and/or Cry2.

## Genetic disruption of Cry1/2 alters the transcriptional response to DNA damage

Given that Cry1 and Cry2 are transcriptional repressors and that we found a robust regulation of their stability by DNA damage, we asked whether the transcriptional response to DNA damage is altered by genetic disruption of Cry1 or Cry2. By measuring the induction of transcripts activated by DNA damage in fibroblasts (*Kenzelmann Broz et al., 2013*), we found that genetic loss of Cry1 or Cry2 enhances or suppresses, respectively, the induction of *Cdkn1a* (*p21*) by genotoxic stress and alters the dynamic response of other established damage responsive transcripts as well (*Figure 5A–E* and *Figure 5—figure supplement 1*). Although the chromatin association of cryptochromes may be different in different cell types, both Cry1 and Cry2 bind some of these loci in mouse liver (*Koike et al., 2012*) (*Figure 5—figure supplement 2*). In addition, Cry1 and Cry2 bind to chromatin regions near several genes encoding proteins that participate in DNA repair (*Supplementary file 3*). Interestingly, the expression of several of those genes in response to DNA damage is also altered by genetic loss of Cry1 and/or Cry2 (*Figure 5F–I*), suggesting that cryptochromes may modulate the activation of DNA repair in response to damage. The regulation of some transcripts in *Cry1⁻/⁻;Cry2⁻/⁻* cells resembles that in *Cry1⁻/⁻* cells (e.g., *Rrm2b, Gadd45a, p16ink4a*), suggesting that Cry1 is more relevant to their regulation than is Cry2. For other transcripts (e.g., *p21, Puma, Xrcc1*), the response to DNA damage in *Cry1⁻/⁻;Cry2⁻/⁻* cells is closer to the response in *Cry2⁻/⁻* cells suggesting that Cry2 is

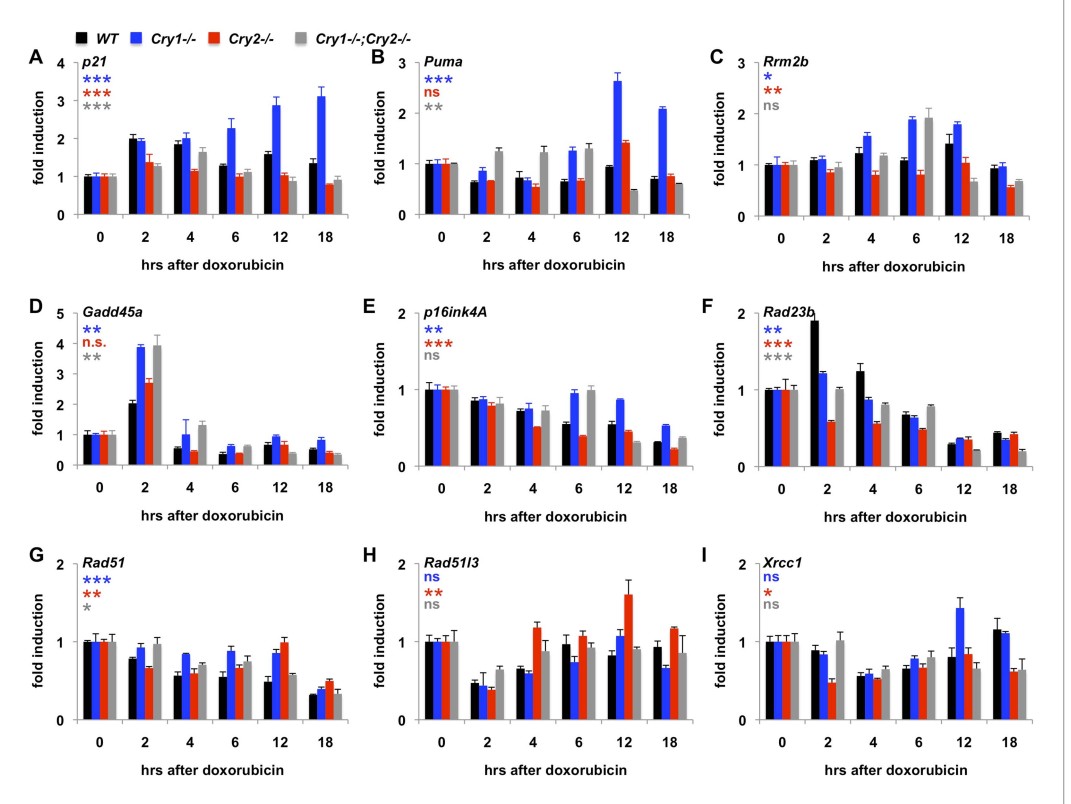

**Figure 5**. Cry1/2 deficiency alters transcriptional response to DNA damage. Expression of the indicated transcripts was measured by quantitative PCR (qPCR) in cDNA from wildtype (black), Cry1$^{-/-}$ (blue), Cry2$^{-/-}$ (red), and Cry1$^{-/-}$; Cry2$^{-/-}$ (gray) fibroblasts treated with doxorubicin for the indicated times. *p < 0.05, **p < 0.01, ***p < 0.001 for effect of genotype by repeated measures ANOVA analysis (blue—WT vs Cry1$^{-/-}$; red—WT vs Cry2$^{-/-}$; gray—WT vs Cry1$^{-/-}$;Cry2$^{-/-}$).

The following figure supplements are available for figure 5:

**Figure supplement 1**. Transcriptional response to irradiation-induced DNA damage.

**Figure supplement 2**. Circadian pattern of Cry1 and Cry2 binding to selected chromatin sites.

more important for regulation of those targets. A full understanding of how Cry1 and Cry2 influence gene expression following DNA damage will require further study.

## Cry2$^{-/-}$ cells accumulate DNA damage

We next asked whether disruption of Cry1/2-dependent transcriptional regulation causes a functional deficit in the cellular response to DNA damage in cryptochrome-deficient cells. Indeed, Cry2$^{-/-}$ and Cry1$^{-/-}$;Cry2$^{-/-}$ fibroblasts accumulate damaged DNA, reflected by an increased percentage of non-dividing cells containing multiple 53BP1-positive foci (*Figure 6A,B*). Accumulation of damaged DNA in cells lacking Cry2 was surprising given that Cry2$^{-/-}$ mice are viable and fertile and that genetic disruption of Cry1 and Cry2 decreases tumor formation in p53-deficient animals (*Ozturk et al., 2009*). To determine whether the increased accumulation of DNA damage that we observed in cells could possibly be relevant in vivo, we analyzed breeding records of a large number of progeny from Cry1$^{+/-}$; Cry2$^{+/-}$ mice over several years: while Cry1 genotypes segregate in the expected Mendelian ratios, the Cry2$^{-/-}$ genotype is significantly underrepresented (*Figure 6C*). This is similar to reduced survival observed in mice with genetic defects in established components of the DNA damage response or DNA repair pathways (*Tsai et al., 2005*; *Mukherjee et al., 2010*; *Crossan et al., 2011*) and is consistent with the possibility that animals lacking Cry2 are prone to genetic instability, though we cannot exclude other possible explanations for the reduced viability of Cry2$^{-/-}$ mice.

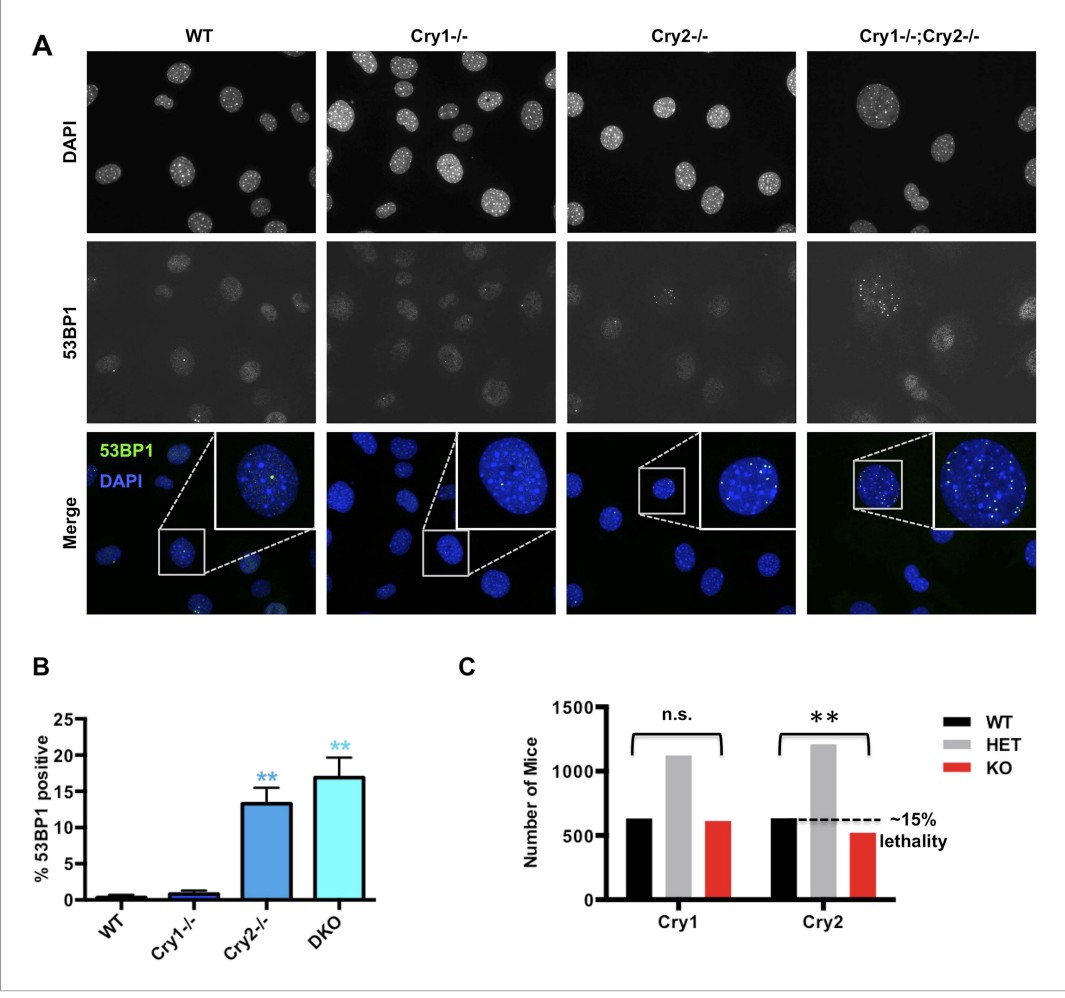

**Figure 6**. *Cry2⁻/⁻* cells accumulate damaged DNA. (**A**) Representative early passage (P3–4) primary wildtype (*WT*), *Cry1⁻/⁻*, *Cry2⁻/⁻*, and *Cry1⁻/⁻;Cry2⁻/⁻* adult ear fibroblasts stained with anti-53BP1 antibody (green) and DAPI (blue). Insets show enlarged view of indicated cells. (**B**) Quantitation of 53BP1-positive cells prepared as described in (**A**). Nuclei containing more than five 53BP1 puncta and negative for BrdU labeling were considered positive for DNA damage. Data represent the mean ± s.d. for at least 200 cells per genotype. (**C**) Chi-squared analysis of the distributions of Cry1 and Cry2 wildtype (black), heterozygous (gray), and homozygous knockout (red) genotypes establishes a significantly reduced survival of *Cry2⁻/⁻* mice. **p < 0.01 by chi-squared analysis with 2 degrees of freedom ($\chi^2 = 10.39$).

## Discussion

Diverse organisms use circadian clocks to optimize the timing of physiological processes in relation to predictable diurnal changes in the external environment (*Dodd et al., 2005*; *Lamia et al., 2008*; *Marcheva et al., 2010*; *Sadacca et al., 2011*). It has long been suspected that clocks influence the timing of cell division to temporally separate DNA replication from predictably recurring exposure to DNA damage (*Sahar and Sassone-Corsi, 2009*; *Sancar et al., 2010*). This hypothesis is supported by the non-random distribution of cell division within circadian cycles (*Nagoshi et al., 2004*). Theoretically, in order for clocks to enable such a separation, their timing must be responsive to genotoxic stress, analogous to entrainment by metabolic signals, which enables clocks to optimally coordinate cellular metabolism with externally fluctuating metabolic demands (*Ramsey and Bass, 2011*; *Jordan and Lamia, 2013*). Indeed, others have shown that DNA damage shifts circadian clock time (*Oklejewicz et al., 2008*). In this study, we confirm that phenomenon and describe a molecular mechanism by which Hausp-mediated deubiquitination and stabilization of Cry1 contributes to it.

Furthermore, we provide evidence for specific and divergent roles of the circadian transcriptional repressors Cry1 and Cry2 in modulating the transcriptional response to DNA damage, thus addressing the longstanding question of whether circadian rhythmicity per se is sufficient to minimize the occurrence or accumulation of DNA damage. Indeed, our results suggest that circadian rhythmicity as such is not protective because *Cry2$^{-/-}$* cells maintain robust circadian rhythms (*Khan et al., 2012*), but they accumulate DNA damage at rates comparable to arrhythmic *Cry1$^{-/-}$;Cry2$^{-/-}$* cells. Therefore, it appears that the genome-protective function of normal circadian rhythms is dependent on the expression of specific clock components, including Cry2. This distinction may underlie some of the controversy over the importance of circadian clocks for maintaining genomic integrity (*Fu and Kettner, 2013*).

Increased accumulation of DNA damage in *Cry2$^{-/-}$* cells would be expected to lead to an increased mutation rate; consistent with that hypothesis, we observed sub-Mendelian inheritance of the *Cry2$^{-/-}$* genotype. Though the potential for circadian clocks to influence the cellular response to DNA damage has been controversial (*Gaddameedhi et al., 2012*), our results are also consistent with several lines of evidence supporting a conserved role for clocks in modulating the DNA damage response and/or DNA repair (*Kang et al., 2009*, *2010*; *Cotta-Ramusino et al., 2011*; *Gaddameedhi et al., 2011*). Interestingly, we identified several proteins that specifically bind damaged DNA or participate in DNA repair in Cry1- and/or Cry2-associated complexes (*Supplementary files 1, 2*), suggesting that cryptochromes may influence DNA repair by non-transcriptional mechanisms as well.

While Cry1 and Cry2 have long been established as repressors of Clock:Bmal1-driven gene expression, and we observe altered expression of several transcripts in response to DNA damage in Cry1/2-deficient cells, it remains unclear how Cry1/2 regulate gene expression. Though it is not the focus of this study, the composition of the Cry1- and Cry2-associated protein complexes suggests that regulation of mRNA processing may be an important function for Cry1 and Cry2: a large number of RNA binding and RNA processing factors were found associated with Cry1 and Cry2 (*Figure 4—figure supplement 2*; *Supplementary files 1, 2*). This is consistent with other recent literature describing the association of RNA processing machinery in complex with Per proteins (*Padmanabhan et al., 2012*) and the importance of post-transcriptional regulation in circadian clock function generally (*Kojima et al., 2011*).

It has long been hypothesized that the C-termini of cryptochromes are important for distinguishing their species-specific functions (*Green, 2004*) and for enabling regulated interactions with protein and possible nucleic acid partners (*Czarna et al., 2011*; *Zoltowski et al., 2011*; *Engelen et al., 2013*). Here, we identify a Cry1-specific partner (Hausp) that interacts with the C-terminus in isolation. Furthermore, we describe phosphorylation events in Cry1, Cry2, and Hausp that are influenced by DNA damage and contribute to their regulated interactions. It appears that DNA damage initiates a complex cascade of signal transduction that alters circadian clock dynamics in a complicated manner. The large number of phosphorylation events in Cry1, Cry2, and Hausp induced by genotoxic stress supports an important role for these proteins in sensing or responding to such stress. Understanding the specific functions of each of these modifications will require further study.

These results suggest that mammalian cryptochromes, Cry1 and Cry2, represent a node of interaction between circadian clocks and the cellular response to genotoxic stress. Cry1 and Cry2 play non-redundant roles in this pathway, highlighting the importance of analyzing their roles independently rather than relying on the exclusive use of doubly deficient cells and animals to understand their functions. Further study will determine whether Cry1 and Cry2 modulate the transcriptional response to DNA damage via sequence-specific DNA binding transcription factors or through direct binding of damaged DNA as has been observed in vitro (*Ozgur and Sancar, 2003*). Finally, accumulation of damaged DNA in *Cry2$^{-/-}$* cells suggests that Cry2 is a particularly important integrator of circadian rhythms and genomic integrity. Taking all of these data together, we conclude that Cry1 and Cry2 cooperatively regulate the transcriptional response to genotoxic stress and the inverse relationship of their stability in response to DNA damage contributes to transient activation of gene networks that protect genome integrity (*Figure 7*).

## Materials and methods

### Mass spectrometry

For mass spectrometry analysis, samples were denatured, reduced, and alkylated before an overnight digestion with trypsin. Peptide mixtures were analyzed by liquid chromatography mass spectrometry

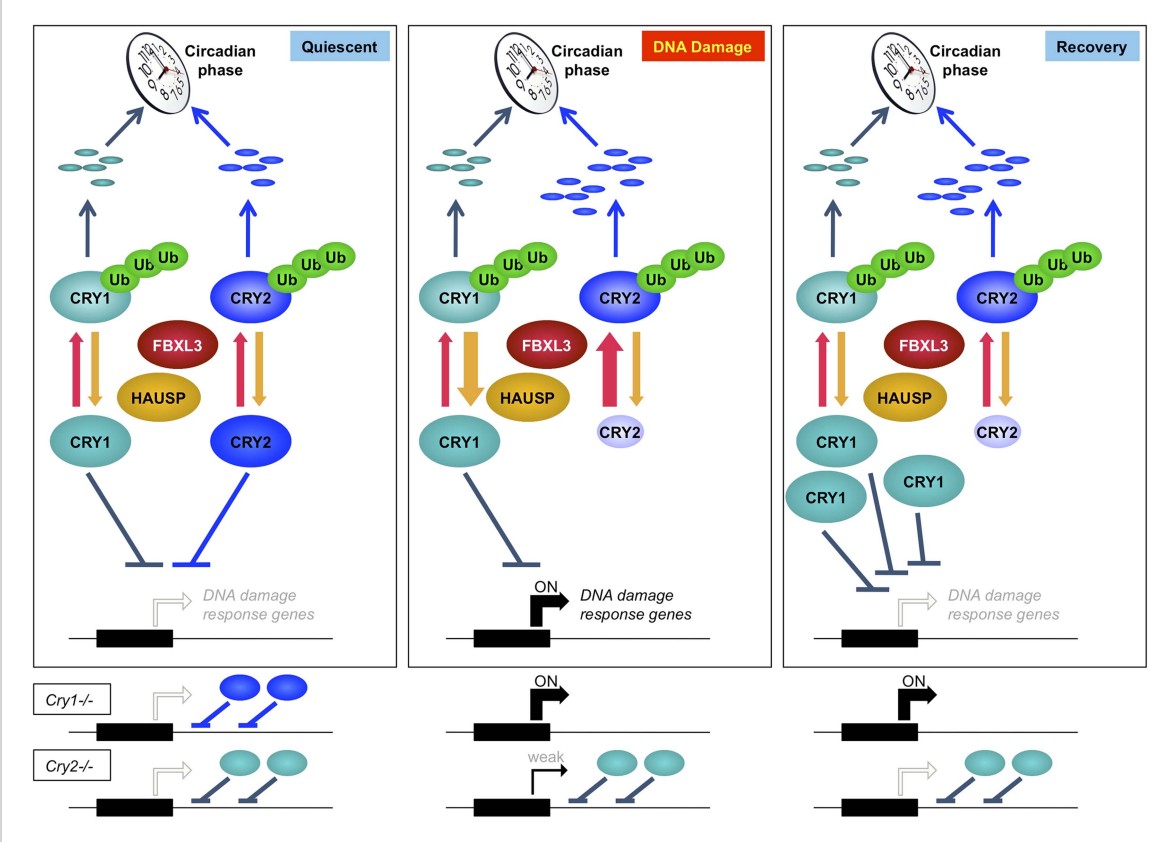

**Figure 7**. Model depicting a novel mechanism by which the regulation of Cry1 and Cry2 enables coordination of the transcriptional response to genotoxic stress. In quiescent cells, Cry1 and Cry2 repress transcription of target genes. Upon DNA damage, Cry2 is degraded, relieving repression. As Cry1 accumulates, it replaces Cry2 and returns gene expression to normal levels resulting in transient activation. In Cry1$^{-/-}$ cells, gene expression is enhanced, while in Cry2$^{-/-}$ cells, damage-induced transcription is suppressed. Note that this model does not explain the dynamics of altered response observed for all transcripts but may apply to the average change in the transcriptional response to DNA damage in Cry1/2-deficient cells.

using an Accela pump and LTQ mass spectrometer (Thermo Fisher Scientific, Waltham, MA) using a four-step multidimensional protein identification technology (MudPIT) separation (*MacCoss et al., 2002*). Tandem mass spectrometry spectra were collected in a data-dependent fashion and resulting spectra were extracted using RawXtract. Protein identification was done with Integrated Proteomics Pipeline (IP2) by searching against the UniProt Human database and filtering to <1% false positive at the protein level using DTASelect. Statistically enriched partners for Cry1 were identified by Pattern Lab (*Carvalho et al., 2012*).

## Cell culture and transfection

All cells were grown in complete Dulbecco's Modified Eagle Medium (DMEM) (cat #10569; Invitrogen) unless otherwise indicated. 293T and U2OSB6 cell media were supplemented with 10% fetal bovine serum, and 1% penicillin and streptomycin. MEF media were supplemented with 15% fetal bovine serum, and 1% penicillin and streptomycin. 293T cells were grown in a 37°C incubator maintained at 5% $CO_2$ and 20% $O_2$ (high oxygen). MEF cells were grown in high oxygen conditions as above (*Figure 2C,D,F*) or a 37°C incubator maintained at 5% $CO_2$ and 3% $O_2$ (low oxygen). Note that in most cases, the MEFs were initially cultured in high oxygen at the time of harvest even when they were later grown in low oxygen. MG-132 was used at a concentration of 10 µM for 6–8 hr or overnight as indicated. AICAR (cat #A61170010; Toronto Research Chemicals, Canada) was used at a concentration of 1–2 mM as indicated. Hausp inhibitor Compound 7 (Progenra, Malvern, PA) was used at a concentration of 10 µM for 6–8 hr prior to cell lysis or as indicated. Doxorubicin (cat #ICN15910101; Thermo Fisher Scientific) was used at a concentration of 0.5 µg/ml for 16–24 hr or as indicated.

Ionizing radiation exposure was performed using a 137Cs γ-radiation source at the indicated times after dexamethasone synchronization. Transfections were carried out using calcium phosphate or polyethylenimine (cat #23966-2; PEI; Polysciences Inc, Warrington, PA) by standard protocols.

## Plasmids and shRNA

pcDNA3-2xFlag-mCRY1, pcDNA3-2xFlag-mCRY2, and pcDNA3-Fbxl3-v5 are as described previously (*Lamia et al., 2009*). pCl-neo Flag HAUSP deposited by Dr Bert Vogelstein was purchased from Addgene (Addgene plasmid 16655) (*Cummins and Vogelstein, 2004*) and cloned into pcDNA 3.2/V5/GW-CAT purchased from Invitrogen (cat #K244020) using standard protocols. Lentiviruses expressing Bmal1-luciferase and Per2-luciferase were from Dr Satchidananda Panda. Five shRNAs against Hausp and one shRNA against Gapdh were purchased from Open Biosystems. pLKO.1 sh_scramble deposited by Dr David Sabatini was purchased from Addgene (Addgene plasmid 1864) (*Sarbassov et al., 2005*). Either sh_Scramble or sh_Gapdh was used as controls for sh_Hausp. pLenti-lox-GFP shRNA p19-2 for immortalizations deposited by Dr Tyler Jacks was purchased from Addgene (Addgene plasmid 14091) (*Sage et al., 2003*). psPAX plasmid (Addgene plasmid 12,260) and pMD2.G plasmid (Addgene plasmid 12259) deposited by Dr Didier Trono used for infection also purchased from Addgene. Cry hybrid constructs were a gift from Dr Andrew C Liu (*Khan et al., 2012*); the hybrid coding sequences were transferred to pcDNA3-2xFlag using standard protocols; several observed mutations in the hybrid coding sequences were corrected by site-directed mutagenesis. All mutations were generated using Agilent Site-Directed Mutagenesis kit and protocols (cat #200521).

## Cell lines

HEK 293T cells were from the American Type Culture Collection (ATCC, Manassas, VA). U2OS-B6 cells were a gift from Dr Satchidananda Panda. MEFs were isolated from embryos of the indicated genotypes at E15.5 and were used as primary (passaged no more than 10 times and grown in 3% oxygen), immortalized with pLenti-lox-GFP shRNA p19-2, or spontaneously immortalized. Ear Fibroblasts were isolated from 3-month-old littermates. Ear punches were put in 70% ethanol for 2 min, washed in PBS, cut into small pieces using a scalpel and transferred to a 15-ml tube. 2 ml of trypsin 0.25% was added and samples were incubated for 1 hr at 37°C in a water bath, vortexing briefly every 10 min. The trypsin was inactivated with 8 ml of EF media (DMEM/15%FBS/PS1%). Cells were spun down 5 min at 1000 rpm, re-suspended in 3 ml of EF media and transferred into a well of a 6-well plate. The medium was changed the next day. Fibroblasts grew after 3–5 days.

## Generation of viruses and stable cell lines

Lentiviral shRNA were transfected into HEK 293T cells using psPAX and pMD2.G packaging plasmids for virus generation. Viral supernatants were collected 48 hr after transfection, filtered through a 0.45-µm filter, supplemented with 6 µg/ml polybrene and added to parental cell lines. After 4 hr, additional media were added to dilute the polybrene to <3 µg/ml. 48 hr after viral transduction, the infected cells were split into selection media containing 5 µg/ml puromycin. Selection media were replaced every 2–3 days until selection was complete (as judged by death of mock-infected cells; typically 1–2 weeks).

## Immunoprecipitation, western blotting, and immunostaining

293T whole cell extracts and mouse liver lysates were prepared using Lysis buffer containing 1%TX-100 as previously described (*Lamia et al., 2004*). MEF cell extracts were prepared from RIPA buffer containing 1% TX-100, 147 mM NaCl, 12 mM sodium deoxycolate, 0.1% SDS, 50 mM Tris pH 8.0, 10 mM EDTA, 50 µM PMSF, phosphatase inhibitors (cat #P5266 and cat #P0044; Sigma, St. Louis, MO) and protease inhibitor (cat #11697498001; Roche, Switzerland). For ubiquitination experiments, iodoacetamide was added to the buffer to a final concentration of 5 mM (Fisher AC122270050).

Antibodies were anti-Flag M2 agarose beads, anti-Flag polyclonal, anti-v5 polyclonal, anti-Lamin A, anti-aTubulin, and anti-βactin from Sigma, anti-Hausp and anti-V5 from Bethyl Labs (Montgomery, TX, cat #A300-033A and cat #A190-120A), anti-53BP1 from Novus Biologicals (Littleton, CO, NB100-304), Cry1-CT and Cry2-CT as described (*Lamia et al., 2011*), anti-p21 from Santa Cruz Biotechnologies (Dallas, TX, cat #sc-6246), anti-p53 as previously described (*Pasini et al., 2004*), and anti-Ubiquitin, anti-phospho-P53 (S15), and anti-phosphoATM/ATR substrate (phospho-SQ/TQ) from Cell Signaling

Technology (Danvers, MA). Anti-Cry1-phosphoS588 antibody was affinity purified from rabbit antisera raised against a phospho-S588 containing peptide.

## Immunofluorescence

Cells were grown on glass coverslips and pulse-labeled for 30 min by adding 10 µM of BrdU to the cell culture medium, washed three times with PBS before fixation with 4% (wt/vol) paraformaldehyde in PBS for 15 min at room temperature (RT) and permeabilized with 0.5% (vol/vol) Triton X-100 in PBS for 10 min at RT. Coverslips were blocked with 1% BSA in PBS for 30 min at RT. For BrdU co-staining, cells were subjected to a DNase I treatment (Sigma; 200 U/ml in 30 mM Tris HCl pH 8.1, 0.33 mM $MgCl_2$, 0.5 mM Mercaptoethanol, 1% BSA, and 0.5% Glycerol) for 1 hr at 37°C in the presence of anti-BrdU 1/50 (BD Pharmingen). Then, coverslips were washed three times with PBS prior to incubation with primary antibodies (anti-53BP1; 1/3000) for 2 hr at RT in blocking buffer. Cells were washed with PBS and incubated with secondary antibodies (Alexa Fluor 488 goat anti-rabbit 1/150 and Alexa Fluor 594 goat anti-mouse 1/150) for 1 hr at RT in blocking buffer. Cells were then washed three times with PBS and stained 15 min with DAPI (0.4 µg/ml in PBS1X) to visualize DNA. The coverslips were mounted onto glass slides with Fluoromount G (Electron microscopy Science). For quantification, at least 200 cells were counted following IF analysis. Cells with at least five 53BP1 foci and negative for BrdU labeling were considered positive for DNA damage. Images were processed using Image J software.

## Nuclear and cytoplasmic fractionation of cultured cells

Cells were washed once with ice cold PBS, fresh cold PBS was added and the cells were transferred to a 5-ml tube and centrifuged 5 min at 2000 rpm. The resulting pellets were washed with cold PBS and transferred to 1.5-ml eppendorf tubes and centrifuged 5 min at 2000 rpm. The resulting pellets were resuspended in Solution A (10 mM Hepes pH 8, 1.5 mM $MgCl_2$, 10 mM KCl, plus protease and phosphatase inhibitors), and incubated for 15 min at 4°C. An equal volume of Solution B (solution A + 1% NP40) was added and the samples were further incubated for 5 min at 4°C and centrifuged 5 min at 3000 rpm. Supernatants from this step represent the cytoplasmic fraction. The remaining nuclear pellets were then washed twice with cold PBS, lysed in RIPA buffer and either used directly (nuclear lysates) or diluted sixfold into IP buffer for immunoprecipitation.

## Lumicycle analysis of circadian period and phase shifts

U2OSB6 cells, MEFs, or adult ear fibroblasts were plated at 100% confluency in 35-mm dishes (cat #82050-538; VWR, Radnor, PA). The next day, cells were treated for 1–2 hr in normal growth medium containing 1 mM dexamethasone and 100 µM D-luciferin. Media were removed and replaced with media containing DMEM, 5% FBS, 1% penicillin-streptomycin, 15 mM Hepes, pH 7.6, and 100 µM D-luciferin. Plates were sealed with vacuum grease (Dow Corning high vacuum grease; cat #59344-055; VWR) and glass cover slips (cat #22038999; 40CIR-1, Fisher Scientific) and placed into the Lumicycle 32 from Actimetrics, Inc. (Wilmette, IL). Data were recorded using Actimetrics Lumicycle Data Collection software and analyzed using Actimetrics Lumicycle Analysis program. Background subtraction of the recorded data was performed with Running Average setting, and fit by least mean squares calculation to a damped sine wave to calculate the period, amplitude, and phase of the curves. Only data with a goodness of fit percentage of 80 or above was included in the analysis.

## Quantitative RT-PCR

RNA was extracted from mouse tissues or cells with Qiazol reagent using standard protocols (cat #799306; Qiagen, Germany). cDNA was prepared using QScript cDNA Supermix (cat #101414-106; VWR) and analyzed for gene expression using quantitative real-time PCR with iQ SYBR Green Supermix (cat #1708885; Biorad, Hercules, CA). For analysis of transcriptional response to DNA damage (doxorubicin or irradiation), cells were used at approximately 70% confluency (*Table 1*).

## In vitro deubiquitination assay

293T cells transiently expressing Flag-tagged Cry1 were treated with 10 µM MG132 for 18 hr and lysed in RIPA buffer containing Roche complete protease inhibitors, 1 mg/ml iodoacetamide, and 50 µM PMSF. Flag-Cry1 was immunoprecipitated for 2 hr with M2-agarose (Sigma A2220), washed five times in RIPA buffer and three times in reaction buffer, and eluted for 1 hr in reaction buffer (60 mM Hepes pH 7.4, 5 mM $MgCl_2$, 4% glycerol, 2 µg/ml aprotinin, 50 µM PMSF, 2 mg/ml BSA) containing 3XFLAG peptide.

**Table 1**. Primers used for qPCR

| | | |
|---|---|---|
| *Cdkn2a (p21)*: | Fwd: CCAGGCCAAGATGGTGTCTT | Rev: TGAGAAAGGATCAGCCATTGC |
| *Mdm2*: | Fwd: CTGTGTCTACCGAGGGTGCT | Rev: CGCTCCAACGGACTTTAACA |
| *Rrm2b*: | Fwd: GACAGCAGAGGAGGTTGACTTG | Rev: AAAACGCTCCACCAAGTTTTCA |
| *Puma*: | Fwd: GTACGGGCGGCGGAGACGAG | Rev: GCACCTAGTTGGGCTCCATTTCTG |
| *Gadd45a*: | Fwd: AAGACCGAAAGGATGGACACG | Rev: CAGGCACAGTACCACGTTATC |
| *Rad23b*: | Fwd: ACCTTCAAGATCGACATCGACC | Rev: ACTTCTGACCTGCTACCGGAA |
| *Rad51l3*: | Fwd: GGAGCTTTGTGCCCAGTACC | Rev: TCCCCAATGTCCCAATGTCTAT |
| *Xrcc1*: | Fwd: AGCCAGGACTCGACCCATT | Rev: CCTTCTCCAACTGTAGGACCA |
| *p16ink4a*: | Fwd: GTGTGCATGACGTGCGGG | Rev: GCAGTTCGAATCTGCACCGTAG |
| *Rad51*: | Fwd: TCACCAGCGCCGGTCAGAGA | Rev: CCGGCCTAAAGGTGCCCTCG |

Equal volumes of eluted Flag-Cry1 were combined with the indicated amounts of recombinant Hausp (cat #E-519; USP7, Boston Biochem) or USP8 (cat #E-520; Boston Biochem, Cambridge, MA) and the reactions were incubated for 30 min at 30°C before adding SDS sample buffer and boiling for 5 min. The resulting samples were separated by 8% SDS-PAGE and Cry1 was detected by immunoblot.

## Mice

$Cry1^{-/-};Cry2^{-/-}$ mice were from Dr Aziz Sancar (*Thresher et al., 1998*); Per2::Luciferase mice (*Yoo et al., 2004*) were purchased from Jackson laboratories (Bar Harbor, ME). All animal care and treatments were in accordance with The Scripps Research Institute guidelines for the care and use of animals under protocol #10-0019.

## Acknowledgements

We thank Drs Andrew Liu (University of Memphis), Satchidananda Panda (The Salk Institute), Eros Lazzerini Denchi (The Scripps Research Institute), and Ben Nicholson (Progenra, Inc.) for providing materials and reagents and Drs Eros Lazzerini Denchi, Supriya Srinivasan, Reuben Shaw, and Joseph Bass for helpful discussions and critical reading of the manuscript. KAL, SJP, ALH, AK, SDJ, and MN were supported by the Searle Scholars Fund, the Sidney Kimmel Foundation for Cancer Research, the Lung Cancer Research Foundation, the National Institute of Diabetes, Digestive and Kidney Diseases (K01DK090188-03 and R01DK097164-01), and a research fellowship from the Deutsche Forschungsgemeinschaft (DFG, to SDJ). JJM and JRY were supported by the National Institute of General Medical Sciences (8 P41 GM103533) and the National Institute on Aging (R01AG027463).

## Additional information

### Funding

| Funder | Grant reference number | Author |
|---|---|---|
| National Institutes of Health (NIH) | K01, DK090188 | Katja A Lamia |
| Kinship Foundation | Searle Scholars Award | Katja A Lamia |
| Sidney Kimmel Foundation for Cancer Research | Cancer Scholar Award | Katja A Lamia |
| Lung Cancer Research Foundation (LCRF) | research project grant | Katja A Lamia |
| National Institutes of Health (NIH) | R01, DK097164 | Katja A Lamia |
| National Institutes of Health (NIH) | R01, AG027463 | YatesJohn R III |

| Funder | Grant reference number | Author |
|---|---|---|
| National Institutes of Health (NIH) | P41, GM103533 | YatesJohn R III |
| Deutsche Forschungsgemeinschaft | research fellowship | Sabine D Jordan |

The funders had no role in study design, data collection and interpretation, or the decision to submit the work for publication.

### Author contributions

SJP, A-LH, SDJ, KAL, Conception and design, Acquisition of data, Analysis and interpretation of data, Drafting or revising the article; AK, Acquisition of data, Analysis and interpretation of data, Contributed unpublished essential data or reagents; MN, JJM, JRY, Acquisition of data, Analysis and interpretation of data

### Author ORCIDs

Sabine D Jordan, http://orcid.org/0000-0001-8974-8522

### Ethics

Animal experimentation: This study was performed in strict accordance with the recommendations in the Guide for the Care and Use of Laboratory Animals of the National Institutes of Health. All of the animals were handled according to approved institutional animal care and use committee (IACUC) protocols (#10-0019) of The Scripps Research Institute.

## Additional files

### Supplementary files

• Supplementary file 1.  Cry1-associated proteins. Lysates from 293T cells expressing pcDNA3-2xFLAG with no insert (control) or Cry1 after the FLAG tag were used to purify control or Cry1-containing complexes by immunoprecipitation (IP) of the FLAG tag. Components of the resulting complexes were identified by mass spectrometry. The experiment was performed in triplicate and PatternLab for Proteomics (Carvalho et al.) was used to identify statistically enriched partners in Cry1-associated complexes compared to the control. Enrichment (Cry1/control) is the ratio of spectral counts in Cry1 vs control samples for all statistically enriched partners over three experiments.

• Supplementary file 2.  Cry2-associated proteins. Lysates from 293T cells expressing pcDNA3-2xFLAG with no insert (control) or Cry2 after the FLAG tag were used to purify control or Cry1-containing complexes by immunoprecipitation (IP) of the FLAG tag. Components of the resulting complexes were identified by mass spectrometry. The experiment was performed in triplicate and PatternLab for Proteomics (Carvalho et al.) was used to identify statistically enriched partners in Cry2-associated complexes compared to the control. Enrichment (Cry2/control) is the ratio of spectral counts in Cry2 vs control samples for all statistically enriched partners over three experiments.

• Supplementary file 3.  Chromatin binding of circadian transcription factors to loci encoding DNA repair proteins. Published data (Koike et al., 2012, Table S2) was searched for the text string 'repair' to make a preliminary identification of chromatin regions near genes involved in DNA repair that were found to be associated with each of the seven circadian transcription factors Cry1, Cry2, Per1, Per2, Clock, Npas2, and Bmal1.

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
