## [Decision Letter]

Thank you for sending your work entitled “Circadian clocks protect genome integrity via Cry1/2-coordinated transcription” for consideration at *eLife*. Your article has been favorably evaluated by Janet Rossant (Senior editor) and 3 reviewers, one of whom, Joseph S Takahashi, served as the guest Reviewing editor.

The Reviewing editor and the other reviewers discussed their comments before we reached this decision, and the Reviewing editor has assembled the following comments to help you prepare a revised submission.

In a proteomic screen for Cry1 interactors, Papp and colleagues find Usp7 (Hausp), and show that the interaction is mediated by the c-terminus of Cry1. In the first part of the manuscript, the authors detail their discovery of a DNA damage-regulated interaction of Cry1 with the deubiquitinase, Hausp. They show that this interaction is specific for Cry1, mediated by its unique C-terminus, and that DNA damage-induced phosphorylation of SQ/TQ sites in the Cry1 C-terminus play a role in assembling the complex. Stabilization of Cry1 after deubiquitination by Hausp occurs only in the nucleus. Importantly, they demonstrate a reciprocal relationship between the damage-induced interaction with Hausp (stabilizing) and interaction with the E3 ubiquitin ligase, Fbxl3 (destabilizing). In the absence of stabilizing Hausp-interactions with Cry2, the authors show that Cry2 undergoes a net destabilization after DNA damage. The authors present data that Hausp is required for DNA damage-dependent phase shifting of the clock, presumably through regulation of Cry1 protein levels. These data are consistent with findings from Oklejewicz et al. that DNA damage induces circadian phase advances through a transcription-independent mechanism. Together, these data provide one of the first molecular descriptions of how DNA damage impacts the stability of circadian proteins.

Comments to be addressed:

1a) Depletion of Usp7 leads to decreased Cry1 expression. However, it also leads to long period length. In contrast, loss of Cry1 by RNAi or mouse knockouts leads to shortened period length, while loss of Fbxl3, which targets Crys for degradation, leads to long period. What explains this apparent discrepancy? In addition, the small molecule KL001 binds directly to Cry1/2 and blocks interaction with Fbxl3 to stabilize Cry1/2 proteins, yet also elicits a long period phenotype (Hirota, Science, 2012, St. John, PNAS, 2014). The authors should strive to be more inclusive of data that illustrate differential effects of Cry1 stabilization on period.

1b) Genotoxic stress stabilizes Cry1 and destabilizes Cry2, increasing the Cry1/Cry2 ratio. However, *Cry1*^*-/-*^ cells (Cry1/Cry2 ratio of 0), have increased responses to genotoxic stress, while *Cry2*^*-/-*^ cells have blunted responses to genotoxic stress. As E-box repressors, both genes have the capacity to repress each other. Is it possible that Cry2 is up-regulated in *Cry1*^*-/-*^ cells and plays the dominant role in responses to genotoxic stress, possibly through its unique c-terminus? Along these lines, are Cry1 and Cr2 differentially recruited to the promoters of DNA damage response genes in Figure 5 in Koike et al.?

2) Figure 4 shows nicely that DNA damage induces interaction of Cry1/2 (Cry2 more noticeably than Cry1) with Fbxl3. However, the experiments probing the role of DNA-damage-dependent phosphorylation of Cry2 T363 or Fbxl3 S388 on their interaction is much less convincing than other data presented in Figure 4. In the absence of data that provide a clearer picture of how DNA damage-dependent phosphorylation regulates this interaction, we suggest removing panels F and G from Figure 4 and their discussion from the manuscript. We do not feel that removing these data detracts from the major findings of the work.

3) In the second part of the manuscript (Figures 5 and 6), the authors look to extend their description of the damage-dependent regulation of Cry1/2 stability to downstream events in cells. Here, the data for Cry1/2 regulation is less convincingly direct. Some genes, like p21, show striking Cry1 or Cry2-dependent changes from WT of opposite magnitude, while others show only a Cry1-dependent change in magnitude or kinetics of induction (Mdm2, *Puma*). It is unclear whether statistical significance should be determined by comparing damage-dependent transcription to the zero point (i.e. without damage) or between genotypes, if we are to assess the role of Cry1/2 in the responses. Moreover, it is not clear from the data presented in Figure 5 how direct the role of Cry1 or Cry2 is in transcriptional regulation of these genes. The authors are admittedly careful not to attribute changes in mRNA to direct regulation by Cry1/2, although it feels as though they imply this direct regulation by citing the Koike et al. findings that Cry1/2 are recruited to sites throughout chromatin. It is unclear what 'Cry1/2-coordinated responses' means in this context and should be better clarified.

4) Finally, in Figure 6, the authors make an intriguing connection between knockout of Cry2 and the accumulation of DNA damage that may play a role in altering Mendelian ratios of survival in Cry2^-/-^ mice. While these data are interesting, we do not see a clear connection between Figures 1, 2, 3 and 4 and Figure 6. If anything, the DNA damage-dependent mRNA profiles shown in Figure 5 seem to suggest that Cry2^-/-^ cells have a transcription response that is not altogether different from WT cells. In addition, Mendelian transmission ratio distortion cannot be directly attributed to “increased mutation rate” without supporting evidence. Many other scenarios are possible. The most obvious would be a defect due to a change in CRY2 targets or function.

5) In Figure 2 and in the fourth paragraph of the Results section, ubiquitination is measured only in cells, therefore, the ubiquitination cannot be assessed “directly” as stated since this requires in vitro ubiquitination assays using recombinant proteins as shown for CRY by [1], in Science and [65], in Cell. The deubiquitination assays should be performed in vitro. It is curious that the Methods section describes a form of “in vitro deubiquitation assay” but this is not what is shown in Figure 2 did not find such results elsewhere. The assay describes the use of immunoprecipitated CRY1 from 293T cells which is not ideal since other interacting CRY1 protein would be co-IP'd. In vitro SCF E3 ligase assays are normally performed using baculovirus expressed CRY constructs and E3 ligase components.

6) In Figure 2, the rightmost two lanes for *Cry*^*-/-*^ there is a (Ub)N signal on the western blot that is as strong as the some of the wildtype lanes on the left. What accounts for this background signal? This emphasizes the need for in vitro ubiquitination assays.

7) Introduction section, first paragraph: The original references for FBXL3 should be cited instead of the Gatfield and Schibler paper (the FBXL21 original papers are cited correctly). [1], Science; [65], Cell; [18], Science; [65].

8) Introduction section, second paragraph: In the list of physiological effects of Cry1/2 null alleles, the response to genotoxic stress induced by cyclophosphamide should be mentioned ([19], PNAS) since, in this case, Cry1/2 knockout mice are resistant to this agent, which on the surface is the opposite of the results reported in this manuscript. In the case of cyclophosphamide, it is a CLOCK;BMAL1 target that confers circadian time-dependent resistance to cyclophosphamide. In addition, a role for CRY has been shown for p53 cancer risk by [54], PNAS.

9) Figure 1: What is responsible for the weak interaction signal for FLAG-N constructs 2-5?

10) Figure 2: Why is this experiment only 2.5 days in length? Too short, should be at least 5 cycles.

---

## [Author Response]

*1a) Depletion of Usp7 leads to decreased Cry1 expression. However, it also leads to long period length. In contrast, loss of Cry1 by RNAi or mouse knockouts leads to shortened period length, while loss of Fbxl3, which targets Crys for degradation, leads to long period. What explains this apparent discrepancy? In addition, the small molecule KL001 binds directly to Cry1/2 and blocks interaction with Fbxl3 to stabilize Cry1/2 proteins, yet also elicits a long period phenotype (*[25]*, St. John, PNAS, 2014). The authors should strive to be more inclusive of data that illustrate differential effects of Cry1 stabilization on period*.

We thank the reviewers for their insight into the complexity of the relationship between Cry1/2 stability and circadian period length and we apologize for the oversimplified discussion of this subject in our original manuscript. Indeed, the mechanism by which Cry1/2 expression or stability is altered influences the ultimate effects on circadian period length. There are several examples in which a change in the average level of Cry1 or Cry2 protein expression can either increase or decrease circadian period depending on the details, some of which are cited by the reviewers. For example, KL001 increases period even in *Cry1*^-/-^ cells (25) showing that stabilization of Cry2 leads to long period length even though knockout or knockdown of Cry2 also increases circadian period. Mathematical modeling spurred the authors of that study to hypothesize that Cry1 and Cry2 have similar roles in period determination and that nuclear stabilization of either isoform leads to period lengthening. We find that destabilization of nuclear Cry1 by shRNA-mediated knockdown of Hausp or pharmacological inhibition of Hausp increases period, which is consistent with the long period in cells treated with AMPK activators that also destabilize nuclear Cry1 but in contrast to short periods observed in *Cry1*^-/-^ cells and long periods in KL001-treated cells in which nuclear Cry1 is stabilized. This discrepancy could be attributed to temporal segregation of the actions of Fbxl3 and Hausp, perhaps one of them is more important during the phase of Cry1/2 accumulation and the other critical during the destruction phase. Alternatively, while the inhibition of Fbxl3 binding and ubiquitylation of Cry1 by KL001 and the acceleration of Cry1 ubiquitylation by Hausp knockdown or inhibition have opposing effects on nuclear Cry1 stability, their similar effects on circadian period length could reflect actions of different subsets of Cry1 (e.g. Cry1 with different posttranslational modifications or in different subnuclear locations) or differential effects on Cry2: for example, in *Cry1*^-/-^ cells and in KL001-treated cells, Cry2 is stabilized while in Hausp-depleted cells, Cry2 is unaffected. Updating mathematical models to incorporate new clock protein partners, posttranslational modifications and the effects on period upon interference with them may help to predict the relative importance of new pathways in various phases of circadian negative feedback (St. John, PNAS, 2014). We have updated the text to accommodate these uncertainties about the relationship between Cry1/2 stability and circadian period determination.

*1b) Genotoxic stress stabilizes Cry1 and destabilizes Cry2, increasing the Cry1/Cry2 ratio. However,* Cry1^-/-^
*cells (Cry1/Cry2 ratio of 0), have increased responses to genotoxic stress, while* Cry2^-/-^
*cells have blunted responses to genotoxic stress. As E-box repressors, both genes have the capacity to repress each other. Is it possible that Cry2 is up-regulated in* Cry1^-/-^
*cells and plays the dominant role in responses to genotoxic stress, possibly through its unique c-terminus? Along these lines, are Cry1 and Cr2 differentially recruited to the promoters of DNA damage response genes in*
Figure 5
*in Koike et al*.*?*

Based on the transcriptional responses to DNA damage that we observed in *Cry1*^-/-^ and *Cry2*^-/-^ cells, we hypothesized that Cry1 and Cry2 redundantly repress some sequence-specific DNA binding transcription factor(s) that contribute to induction of transcription in response to genotoxic stress. For example: if Cry2 has a slightly higher affinity than does Cry1 for transcription factor X, in quiescent wildtype cells, X is repressed by Cry2 and poised for activation upon Cry2 degradation following DNA damage. Accumulation of Cry1 after DNA damage would allow Cry1 to replace the degraded Cry2 and return expression of the target genes of X to basal levels resulting in transient activation in wildtype cells while expression of those genes would be sustained in *Cry1*^-/-^ cells as we observed. The activation of those genes would be blunted in *Cry2*^-/-^ cells due to Cry1 being bound to X in the absence of Cry2 since Cry1 would not be degraded in response to the damage stimulus (Figure 7). As the reviewers noted, this pattern of altered regulation was most clear and consistent in the case of *Cdkn1a* (*p21*) expression, but *Rrm2b* also exhibits increased induction in *Cry1*^-/-^ cells and blunted induction in *Cry2*^-/-^ cells (Figure 5) and *Gadd45a* and *Puma* are hyperactivated in *Cry1*^-/-^ cells while the response in *Cry2*^-/-^ cells is suppressed in response to irradiation (Figure 5—figure supplement 1). We did not detect induction of *Gadd45a* or *Puma* in either wildtype or *Cry2*^-/-^ cells in response to doxorubicin treatment under the conditions used, so we can’t say whether their response to doxorubicin would be blunted in *Cry2*^-/-^ cells.

The reviewers correctly point out that a model in which Cry2 positively contributes to activation of target genes by a DNA binding transcription factor X would also explain our data. We had not considered this because Cry1 and Cry2 are generally believed to act as transcriptional repressors and the observed degradation of Cry2 in response to DNA damage could explain activation of associated transcription factors but it is certainly possible that a transiently induced state of Cry2 (for example ubiquitylated Cry2) is required to activate transcription and we thank the reviewers for this creative suggestion.

In order to gain additional insight into the roles of Cry1 and Cry2 in transcriptional regulation after DNA damage, and in particular to determine whether the effects observed in either *Cry1*^-/-^ or *Cry2*^-/-^ cells could be attributed to increased expression of the remaining cryptochrome, we examined transcription in response to DNA damage in *Cry1*^-/-^; *Cry2*^*-/-*^ doubly deficient (*DKO*) fibroblasts in addition to the single knockout cells. We also measured several additional transcripts including some inspired by the reviewers’ suggestion to mine the Koike et al. ChIP data set for possible direct targets of Cry1 and/or Cry2. The results are illuminating and suggest that both Cry1 and Cry2 can participate in the regulation of transcription in response to DNA damage, though their relative importance seems to depend on the specific transcript measured and on the duration of the DNA damage response.

The expression pattern for *p21* in *DKO* cells looks similar to that of *Cry2*^-/-^ cells, though we generally observe at least some induction of *p21* in *DKO* cells and less or none in *Cry2*^-/-^ cells, suggesting that the loss of Cry2 is sufficient to explain some blunted induction of *p21* but that increased Cry1 protein could contribute to the more severe loss of *p21* induction in *Cry2*^-/-^ cells compared to *DKO* cells. In contrast, the prolonged activation of *p21* in response to DNA damage in *Cry1*^-/-^ cells seems to require the expression of Cry2 since it does not occur in the *DKO* cells, suggesting that this phenomenon could involve either release of repression upon Cry2 degradation or a positive action of a subset of Cry2 (e.g. uniquitylated), or an indirect effect of the loss of Cry2 on another pathway that modulates *p21* expression.

The expression patterns of *Gadd45a* and *Rrm2b* in *DKO* cells resemble the patterns in *Cry1*^-/-^ cells, suggesting that it is the loss of Cry1 rather than accumulation of Cry2 that causes hyperactivation of those transcripts. Interestingly, the *Gadd45a* locus was observed bound to both Cry1 and Cry2 and the *Rrm2b* locus was uniquely bound by Cry1 in the Koike et al. data set, suggesting that Cry1-dependent regulation of these genes could involve direct binding (Figure 5—figure supplement 2).

While we are cautious about using ChIP-sequencing data from unstimulated liver samples to interpret the effects of genetic loss of Cry1 and Cry2 on the transcriptional response to DNA damage stimuli in fibroblasts, we examined the association of each of the circadian transcription factors (Koike et al.) not only with the established DNA damage responsive genes that we initially analyzed but also with chromatin regions that encode proteins involved in DNA repair. Interestingly, Cry1 and Cry2 were bound to several more such sites than were the other circadian factors ([Supplementary-material SD3-data]).

In order to assess whether these associations in unstimulated livers are relevant to DNA damage response in fibroblasts, we measured the expression of these transcripts in response to DNA damage in WT and Cry1/2-deficient cells and found that several of their expression profiles are altered by genetic disruption of Cry1 and/or Cry2. Most striking, *Rad23b* expression is induced only in the wildtype cells: disruption of Cry1 and/or especially Cry2 prevents its activation by DNA damage.

These analyses that were inspired by the reviewer’s suggestions indicate that Cry1 and Cry2 can both directly and indirectly modulate the transcriptional response to DNA damage. A more complete understanding of how Cry1 and Cry2 influence gene expression during DNA damage response will require analysis of genome-wide transcription and chromatin association of Cry1 and Cry2 in wildtype and Cry1/2-deficient cells in response to DNA damage, which is beyond the scope of this study. We have updated the text to clarify the state of our understanding and we changed the title of the manuscript to emphasize other aspects of this work that are more central to the novel and important findings described herein.

*2)*
Figure 4
*shows nicely that DNA damage induces interaction of Cry1/2 (Cry2 more noticeably than Cry1) with Fbxl3. However, the experiments probing the role of DNA-damage-dependent phosphorylation of Cry2 T363 or Fbxl3 S388 on their interaction is much less convincing than other data presented in*
Figure 4*. In the absence of data that provide a clearer picture of how DNA damage-dependent phosphorylation regulates this interaction, we suggest removing panels F and G from*
Figure 4
*and their discussion from the manuscript. We do not feel that removing these data detracts from the major findings of the work*.

We agree with the reviewers’ comments and have removed these data from the manuscript.

*3) In the second part of the manuscript (*Figures 5 and 6*), the authors look to extend their description of the damage-dependent regulation of Cry1/2 stability to downstream events in cells. Here, the data for Cry1/2 regulation is less convincingly direct. Some genes, like p21, show striking Cry1 or Cry2-dependent changes from WT of opposite magnitude, while others show only a Cry1-dependent change in magnitude or kinetics of induction (Mdm2,* Puma*). It is unclear whether statistical significance should be determined by comparing damage-dependent transcription to the zero point (i.e. without damage) or between genotypes, if we are to assess the role of Cry1/2 in the responses. Moreover, it is not clear from the data presented in*
Figure 5
*how direct the role of Cry1 or Cry2 is in transcriptional regulation of these genes. The authors are admittedly careful not to attribute changes in mRNA to direct regulation by Cry1/2, although it feels as though they imply this direct regulation by citing the Koike et al. findings that Cry1/2 are recruited to sites throughout chromatin. It is unclear what 'Cry1/2-coordinated responses' means in this context and should be better clarified*.

We agree that we have not thoroughly investigated the roles of Cry1 and Cry2 in the transcriptional response to DNA damage and that the relationship is likely to be complex and transcript dependent as discussed in response to point #1b above. We also agree that a more appropriate statistical analysis should be applied and have now analyzed these data by repeated measures ANOVA to ask whether the Cry1/2 genotype significantly alters the response to the stimulus (Figure 5). As noted by the reviewers we have been careful not to attribute the observed transcriptional changes to direct regulation by Cry1 and/or Cry2 because we are not able to precisely discern the mechanism(s) by which they modulate the transcriptional response to DNA damage. We have clarified the text on these points.

*4) Finally, in*
Figure 6*, the authors make an intriguing connection between knockout of Cry2 and the accumulation of DNA damage that may play a role in altering Mendelian ratios of survival in* Cry2^-/-^
*mice. While these data are interesting, we do not see a clear connection between*
Figures 1, 2, 3 and 4
*and*
Figure 6*. If anything, the DNA damage-dependent mRNA profiles shown in*
Figure 5
*seem to suggest that* Cry2^-/-^
*cells have a transcription response that is not altogether different from WT cells. In addition, Mendelian transmission ratio distortion cannot be directly attributed to* “*increased mutation rate*” *without supporting evidence. Many other scenarios are possible. The most obvious would be a defect due to a change in CRY2 targets or function*.

We agree that many other scenarios could explain the Mendelian ratio distortion in Cry2-deficient animals. We performed this analysis because we were surprised to observe striking accumulation of DNA damage in *Cry2*^*-/-*^ cells and wondered whether this could possibly be relevant in vivo given that *Cry2*^*-/-*^ mice are viable and fertile, while several genetic models of mice with deficiencies in DNA damage response or DNA repair have been shown to be born at sub-Mendelian ratios (68; 47; 4). While we cannot determine the root cause of the Mendelian ratio distortion at this time, it is at least consistent with the possibility that Cry2 is important for maintaining genomic integrity in vivo consistent with the results that we observed in cultured cells. We have updated the text to clarify this.

In addition, we added new quantitative analysis of the proteomics data sets that identified Hausp as a strong partner for Cry1, which also identified several proteins that interact with damaged DNA and/or participate in DNA repair in complex with Cry1 and/or Cry2 ([Supplementary-material SD1-data SD2-data]). We have updated the Discussion to include this information and to better connect the different parts of the paper.

*5) In*
Figure 2
*and in the fourth paragraph of the Results section, ubiquitination is measured only in cells, therefore, the ubiquitination cannot be assessed* “*directly*” *as stated since this requires in vitro ubiquitination assays using recombinant proteins as shown for CRY by Busino, 2007, in Science and Siepka, 2007, in Cell. The deubiquitination assays should be performed in vitro. It is curious that the Methods section describes a form of* “*in vitro deubiquitation assay*” *but this is not what is shown in*
Figure 2
*did not find such results elsewhere. The assay describes the use of immunoprecipitated CRY1 from 293T cells which is not ideal since other interacting CRY1 protein would be co-IP'd. In vitro SCF E3 ligase assays are normally performed using baculovirus expressed CRY constructs and E3 ligase components*.

We have added data (Figure 2—figure supplement 2) demonstrating that purified recombinant HAUSP (USP7) can remove ubiquitin chains from ubiquitinated Cry1 purified from 293T cells treated with the proteasome inhibitor MG132 while the related deubiquitinase USP8 did not remove ubiquitin from Cry1. While immunoprecipitated Cry1 may contain other interacting proteins, the dose-dependent removal of ubiquitin from this substrate by recombinant HAUSP (and not recombinant USP8) added to the reaction in vitro suggests that Hausp can directly catalyze the removal of ubiquitin from Cry1, though it is possible that it activates an associated protein that directly deubiquitinates Cry1. We were not able to use baculovirus expressed Cry1 for this assay because the C-terminal tail of Cry1 is required for its interaction with Hausp and Cry1 can only be stably expressed and purified from baculovirus-infected Sf9 cells if the C-terminal tail is excluded. (Instability of full length Cry2 was described in [72], and we were similarly able to stably express and purify only Cry1 and Cry2 lacking the C termini in baculovirus expression systems).

*6) In*
Figure 2*, the rightmost two lanes for Cry*^*-/-*^
*there is a (Ub)N signal on the western blot that is as strong as the some of the wildtype lanes on the left. What accounts for this background signal? This emphasizes the need for in vitro ubiquitination assays*.

The rightmost two lanes in Figure 2 were included to control for antibody specificity in the immunoprecipitation. The signal in the first lane (WT cells without proteasome inhibition) is similar to that in the control lanes, suggesting that under normal conditions, ubiquitinated Cry1 is rapidly degraded by the proteasome and is not detectable by this assay, at least not above this background signal, which likely represents other ubiquitinated proteins that are non-specifically immunoprecipitated in all of the samples. In the second lane, from WT cells in which proteasome activity is blocked by mg132 treatment, we clearly see an increase in the specific signal for ubiquitinated Cry1. We now include quantitation of these blots subtracting the average signal in the rightmost two lanes from the other lanes to quantitatively assess the changes in Cry1 ubiquitination resulting from Hausp depletion in the presence and absence of proteasome inhibition (Figure 2—figure supplement 3).

*7) Introduction section, first paragraph: The original references for FBXL3 should be cited instead of the Gatfield and Schibler paper (the FBXL21 original papers are cited correctly).*
[1]*, Science;*
[65]*, Cell;*
[18]*, Science;*
[65].

We have corrected these citations.

*8) Introduction section, second paragraph: In the list of physiological effects of Cry1/2 null alleles, the response to genotoxic stress induced by cyclophosphamide should be mentioned (*[19]*, PNAS) since, in this case, Cry1/2 knockout mice are resistant to this agent, which on the surface is the opposite of the results reported in this manuscript. In the case of cyclophosphamide, it is a CLOCK;BMAL1 target that confers circadian time-dependent resistance to cyclophosphamide. In addition, a role for CRY has been shown for p53 cancer risk by*
[54]*, PNAS*.

We have updated the text to include these additional physiological effects of Cry1/2 genetic disruption.

*9)*
Figure 1: *What is responsible for the weak interaction signal for FLAG-N constructs 2-5?*

In cells overexpressing Hausp and Cry2, we observe an interaction between the two proteins, albeit much weaker than that between Hausp and Cry1; similarly, all of the Cry1/2 hybrids interact weakly with Hausp. Those that contain the Cry2 C-terminus (#2-5) interact weakly with Hausp; those that contain the Cry1 C-terminus (#1, #6) bind strongly. We do not believe that the Cry2-Hausp interaction is biologically relevant as we were not able to detect interaction between endogenous Cry2 and Hausp, and Cry2 protein levels were not altered by shRNA knockdown (Figure 3) or pharmacological inhibition of Hausp (not shown).

*10)*
Figure 2*: Why is this experiment only 2.5 days in length? Too short, should be at least 5 cycles*.

We have updated Figure 2 to include 5 cycles.